# Reconsideration of the Appropriate Dissection Range Based on Japanese Anatomical Classification for Resectable Pancreatic Head Cancer in the Era of Multimodal Treatment

**DOI:** 10.3390/cancers13143605

**Published:** 2021-07-19

**Authors:** Yuichi Nagakawa, Naoya Nakagawa, Chie Takishita, Ichiro Uyama, Shingo Kozono, Hiroaki Osakabe, Kenta Suzuki, Nobuhiko Nakagawa, Yuichi Hosokawa, Tomoki Shirota, Masayuki Honda, Tesshi Yamada, Kenji Katsumata, Akihiko Tsuchida

**Affiliations:** 1Department of Gastrointestinal and Pediatric Surgery, Tokyo Medical University, Tokyo 160-8402, Japan; nao8nak@tokyo-med.ac.jp (N.N.); chie0428@tokyo-med.ac.jp (C.T.); s-kozono@tokyo-med.ac.jp (S.K.); osakabeh@tokyo-med.ac.jp (H.O.); ks_0828@tokyo-med.ac.jp (K.S.); nkgw_n1@tokyo-med.ac.jp (N.N.); yhosokaw@tokyo-med.ac.jp (Y.H.); but_ibshigin@tokyo-med.ac.jp (T.S.); masayuki@tokyo-med.ac.jp (M.H.); exk0009@tokyo-med.ac.jp (T.Y.); k-katsu@tokyo-med.ac.jp (K.K.); akihikot@tokyo-med.ac.jp (A.T.); 2Department of Advanced Robotic and Endoscopic Surgery, Fujita Health University, Toyoake 470-1192, Aichi, Japan; iuyama@fujita-hu.ac.jp; 3Division of Chemotherapy and Clinical Research, National Cancer Center Research Institute, Tokyo 104-0045, Japan

**Keywords:** pancreatic cancer, pancreaticoduodenectomy, mesopancreas, superior mesenteric artery, nerve and fibrous tissues, adjuvant chemotherapy, lymph node dissection, R0 resection

## Abstract

**Simple Summary:**

Although the survival benefit of “regional lymph node dissection” for pancreatic head cancer remains unclear, the R0 resection rate is reportedly associated with prognosis. We reviewed the literature that could be helpful in determining the appropriate resection range. The recent development of high-quality computed tomography has made it possible to evaluate the extent of cancer infiltration. Even if the “dissection to achieve R0 resection” range is simulated based on the computed tomography evaluation, it is difficult to identify the range intraoperatively. It is necessary to be aware of the anatomical landmarks to determine the appropriate dissection range intraoperatively.

**Abstract:**

Patients with resectable pancreatic cancer are considered to already have micro-distant metastasis, because most of the recurrence patterns postoperatively are distant metastases. Multimodal treatment dramatically improves prognosis; thus, micro-distant metastasis is considered to be controlled by chemotherapy. The survival benefit of “regional lymph node dissection” for pancreatic head cancer remains unclear. We reviewed the literature that could be helpful in determining the appropriate resection range. Regional lymph nodes with no suspected metastases on preoperative imaging may become areas treated with preoperative and postoperative adjuvant chemotherapy. Many studies have reported that the R0 resection rate is associated with prognosis. Thus, “dissection to achieve R0 resection” is required. The recent development of high-quality computed tomography has made it possible to evaluate the extent of cancer infiltration. Therefore, it is possible to simulate the dissection range to achieve R0 resection preoperatively. However, it is often difficult to distinguish between areas of inflammatory changes and cancer infiltration during resection. Even if the “dissection to achieve R0 resection” range is simulated based on the computed tomography evaluation, it is difficult to identify the range intraoperatively. It is necessary to be aware of anatomical landmarks to determine the appropriate dissection range during surgery.

## 1. Introduction

Pancreatic ductal adenocarcinoma (PDAC) is recognized as having one of the poorest prognoses of all tumors. Resection is the only treatment that can result in long-term survival. Several randomized controlled trials have shown that extended lymph node dissection does not provide survival benefits in patients with pancreatic head cancer, despite a prolonged operative time and increased blood loss [1,2,3,4,5]. Regional lymph node dissection for pancreatic head cancer has been performed in many facilities, but its survival benefit remains unclear. On the other hand, pancreatic cancer treatment has dramatically changed recently owing to the development of effective chemotherapy. Adjuvant chemotherapy is essential for improving the prognosis of pancreatic cancer [6,7]. A randomized prospective study showed that the introduction of preoperative chemotherapy led to a prolonged prognosis in patients with pancreatic cancer [8]. Most of the recurrence patterns of resectable pancreatic cancer are distant metastases, and resectable pancreatic cancer is considered to be a systemic disease with micrometastasis. Thus, multimodal treatment is required to improve the prognosis of resectable pancreatic cancer, and pancreatic resection should be performed with consideration of preoperative and postoperative treatment.

However, many studies reported that the R0 resection rate is associated with prognosis. The recent development of high-quality high-resolution multi-detector computed tomography (MDCT) has made it possible to evaluate the extent of cancer progression, which makes it possible to simulate the appropriate dissection range to achieve R0 surgery before surgery. Even if the dissection range is simulated preoperatively, an accurate understanding of the anatomical structure is required to identify the dissection range during surgery. In the era of multidisciplinary treatment for resectable pancreatic cancer, we reviewed the literature that could be helpful in determining the appropriate resection range.

## 2. Is “Regional LYMPH Node Dissection” Required?

Patients with pancreatic cancer often have lymph node metastasis, and many studies have reported that lymph node metastasis is a prognostic factor [9,10,11]. Prior to the development of effective adjuvant chemotherapy, extended lymph node dissection, including para-aortic lymph nodes, was performed to prevent local recurrence [12,13]. However, several randomized controlled trials have shown that extended lymphadenectomy does not provide survival benefits in patients with pancreatic head cancer, despite a prolonged operative time and increased blood loss [1,2,3,4,5]. However, it has been reported that the number of retrieved lymph nodes is associated with R0 resection rates and survival [14].

The regional lymph nodes are numbered according to the Japanese Pancreatic Cancer classification [15]. Regional lymph nodes for pancreaticoduodenectomy are classified into Group 1 (8a, 8p, 13a, 13b, 17a and 17b) and Group 2 (5.6, 12a, 12b, 12p, 14p and 14d). The lymph node dissection in Group 1 is defined as D1 dissection, and the lymph node dissection of Group 1 and Group 2 is defined as D2 dissection (Figure 1A). However, it was reported that there was no significant difference in prognosis between D1 and D2 dissections in a randomized controlled trial [5] (Table 1), and it is still debated whether prophylactic dissection of regional lymph nodes improves prognosis [16]. Using surgical results of 495 patients with PDAC, Imamura et al. calculated the efficacy index for each lymph node station by multiplying the frequency of lymph node metastasis to the station and survival to clarify the optimal extent of lymph node dissection. Their results indicated that the efficacy of lymph node dissection differs between uncinate process cancer and pancreatic neck cancer, and the extent of dissection should be determined according to the location of the tumor. They also showed that the site of regional lymph node and lymph node recurrence pattern are different, indicating that it may be necessary to reconsider the need for regional lymph node dissection [15].

In the area around the superior mesenteric artery (SMA), “regional lymph node dissection” also dissects the adipose and connective tissues around the regional lymph nodes, which is almost the same dissection range as “dissection to achieve R0 resection.” On the other hand, the 14p, 14d, and 8p lymph nodes, which are located around the CHA and the SMA, cannot be identified during resection because the lymph nodes are covered with many nerves and fibers. These regional lymph nodes may be confused with other numbers of lymph nodes. Thus, it is difficult to identify the precise location of each regional lymph node during surgery. Novel criteria may be needed to determine the appropriate lymph node dissection area [17].

Here, it should be noted that “regional lymph node dissection” and “dissection to achieve R0 resection” have different purposes. “Dissection to achieve R0 resection” is performed to avoid residual cancer infiltration, whereas “regional lymph node dissection” is performed to prevent recurrence of the lymph nodes. Thus, “regional lymph node dissection” and “dissection to achieve R0 resection” should be separately when considering the appropriate dissection range for resectable PDAC. Multimodal treatment, including neoadjuvant therapy and postoperative adjuvant chemotherapy, dramatically improve prognosis. Regional lymph nodes with no suspected metastases on preoperative imaging may become areas treated with preoperative and postoperative adjuvant chemotherapy. Further discussion is needed to clarify the necessity of “regional lymph node dissection.”

## 3. Is “Dissection to Achieve R0 RESECTION” Required?

Many studies have described the need for R0 resection to achieve long-term survival, and the results of most studies have shown that R0 resection improves the survival rate of patients with resectable PDAC who have undergone pancreaticoduodenectomy (PD) [18,19]. Ghaneh et al. [20] analyzed data from the European Study Group for Pancreatic Cancer-3 randomized controlled trial and found that R1 (direct) resections were associated with significantly reduced overall and recurrence-free survival following pancreatic cancer resection. Resection margin involvement was also associated with an increased risk of local recurrence. Based on these results, the National Comprehensive Cancer Network (NCCN) guidelines have described that the goals of surgical extirpation of pancreatic carcinoma focus on the achievement of an R0 resection, as a margin-positive specimen is associated with poor long-term survival. In contrast, Schmocke et al. [21] retrospectively examined 468 patients with resectable pancreatic cancer or borderline resectable pancreatic cancer who received preoperative treatment. They reported that margin status was not a significant predictor of overall survival or relapse-free survival in multivariate analysis, but the clinical stage, duration of *N*-acetyl cysteine treatment, nodal status, histopathologic treatment response score, and receipt of adjuvant chemotherapy were factors associated with overall survival. In contrast, in pancreaticoduodenectomy with a complicated cutting surface, the R0 resection rate may differ depending on the evaluation and slicing methods [18]. Additionally, the inking of the cut surface according to a defined color code leads to an accurate R0/R1 evaluation [22]. Two definitions have been reported in assessing R1 [23]. American and Japanese classifications define R1 as direct microscopic involvement at the resection margin (0 mm rule) [15,24], and the Royal University of Pathologists classification defines R1 as the presence of cancer cells within 1 mm of the resection margin (1 mm rule) [25]. It is still unclear which classification reflects the prognosis [26]. To clarify the need to achieve R0 resection to prolong the prognosis in the era of multimodal treatments, the pathological evaluation should be standardized. Currently, there is little evidence that “R0 resection is not needed” to improve prognosis. Therefore, even in the era of multimodal treatment, resectable pancreatic cancer may require surgery to achieve R0 resection.

## 4. The Issue Regarding Tumor Infiltration of Nerve and Fibrous Tissues

Dense connective tissues exist around the pancreatic head, which is composed of intensive nerve and fibrous tissues (NFTs). It has been reported that dissection of the NFTs around the pancreatic head is important for achieving R0 resection because PDAC often infiltrate these NFTs [27,28]. However, the appropriate dissection range of NFTs has not been fully discussed, and a common classification showing the anatomical structure of NFTs around the pancreatic head is needed to determine the dissection range. Several classifications have been shown for the anatomy of NFTs. The Japanese classification for pancreatic cancer shows the anatomy of NFTs around the pancreatic head. In this classification, the major NFTs connecting to the pancreatic head are classified into two pathways. One is the pathway from the right celiac ganglion to the posterior side of the pancreas head (pancreatic head plexus I; PLph I), and the other is the pathway from the SMA nerve plexus to the left side of the uncinate process (pancreatic head nerve plexus II; PLph II) [15]. Nagakawa et al. [29] classified the intensive NTFs spreading around the SMA into four areas based on the autopsy findings. Area A: NFTs spreading from the right celiac ganglion and the superior side of the pancreatic head and the posterior side of the hepatoduodenal ligament. Area B: NFTs spreading from the SMA nerve plexus and the uncinate process. Area C: NFTs spreading from the SMA nerve plexus to the anterior side of the jejunal mesentery. Area D: NFTs spreading from the inferior side of the uncinate process to the posterior side of the jejunal mesentery. They also found three SMA nerve plexus regions without branching nerves (SMA I-III) and described that these regions become good anatomical landmarks to identify the SMA nerve plexus before stating these NFTs areas. These anatomical classifications may become good criteria for determining the appropriate dissection range of NFTs.

## 5. Determination of the Appropriate Dissection Range

Intraoperative pathological diagnosis using frozen section is generally performed to determine the pancreatic cutting line to avoid positive pancreatic neck margins. Additionally, resectability status is also evaluated using frozen sections of the SMA margin in some facilities. Nirsgke et al. reported that long-term survival was improved by re-resecting the positive surgical margin found using frozen section to achieve R0 resection [30]. However, many studies reported that intraoperative frozen section-based re-resection of R1 margins does not improve overall survival for patients with PDAC [31,32,33].

In patients with pancreatic head cancer, the extent of cancer infiltration varies depending on the tumor position (e.g., the difference between the pancreatic head and uncinate process) [34,35,36,37]. The development of MDCT has made it possible to confirm the accurate infiltration range of pancreatic head cancer, which can simulate the dissection range preoperatively to achieve R0 resection [38,39,40,41]. However, the extent of tumor infiltration cannot be accurately confirmed during surgery. It is often difficult to distinguish between areas of inflammatory changes and cancer infiltration during resection. Even if the “dissection to achieve R0 resection” range is simulated based on the MDCT evaluation, it is difficult to identify the range intraoperatively. Therefore, anatomical structures, such as layers, arteries, and veins, are commonly identified during surgery as anatomical landmarks to determine the dissection region [29,42].

## 6. Anatomical Landmarks Used to Determine the Appropriate Dissection Range at Each Surgical Site

We summarize below the anatomical structures that can be used as landmarks at each surgical site for achieving R0 resection based on preoperative diagnostic imaging.

### 6.1. Dissection around the Hepatoduodenal Ligament and Common Hepatic Artery

The dissection range around the hepatoduodenal mesentery and common hepatic artery (CHA) may need to be altered according to the tumor location. Uncinate process cancer invades the SMA mainly through the second part of the PLph II (equivalent to Area B) [29,34,36,43] (Figure 1B). However, in pancreatic head cancer, infiltration and lymph node metastasis around the CHA and hepatoduodenal ligament are observed [17,36]. There are 8a lymph nodes on the anterior side of the CHA, which must be removed to expose the CHA, proper hepatic artery (PHA), gastroduodenal artery (GDA), and portal vein (PV) at the superior border of the pancreas.

There is a left celiac ganglion on the right side of the root of the CHA and SMA, and nerve and fibrous tissues (NFTs) spread from the left celiac ganglion to the head of the pancreas and hepatoduodenal ligament (Area A, Figure 1C). These NFTs are divided into NFTs (PLph I) that pass through the dorsal side of the GDA (Figure 1B) and toward the upper edge of the head of the pancreas, and NFTs that pass through the dorsal side of the PHA and extend to the hepatoduodenal ligament [44] (Figure 1C). NFTs spreading to the hepatoduodenal ligament include 8p, 12p, and lymph nodes wrapped in adipose tissue [44]. These NFT regions need to be dissected when attempting complete skeletonization of the PV around the hepatic arteries around the hepatoduodenal ligament. If uncinate process cancer infiltrates around the SMA root and exposure of the CHA root is attempted, these NFT regions also need to be dissected. On the other hand, if no tumor extension is observed around the hepatoduodenal ligament and/or SMA root and CHA root, it is anatomically possible to preserve these NTF regions (Figure 2A–E).

### 6.2. Posterior Dissection

Few studies have described the appropriate range of posterior dissection for pancreatic head cancer. In extended lymph node dissection, including the para-aortic lymph nodes, the inferior vena cava, left renal vein, and anterior surface of the aorta are exposed. However, periaortic lymph node metastasis is now categorized as distant metastasis [15]. Prophylactic periaortic lymph node dissection is not generally performed for resectable PDAC. Delpero et al. investigated the association between each margin status and prognosis in a multicenter prospective study of 150 patients who underwent macroscopic margin-free PD. They showed that the R1 rate was 23%, while only 7% had R1 at the posterior margin; in addition, they reported that posterior R1 was not a prognostic factor [45]. Therefore, “dissection to achieve R0 resection” may not be necessary.

There is a fusion fascia between the posterior side of the pancreatic head and the anterior side of the vena cava and the aorta, which is called the fusion fascia of Treitz [46]. There is loose connective tissue at the anterior surface of this fusion fascia, which can be easily peeled off. If posterior infiltration is not found on the preoperative computed tomography image, this fusion fascia becomes a good anatomical landmark for indicating the range of posterior dissection. If posterior infiltration is suspected before resection and dissection with a surgical margin is needed, the anterior surface of the vena cava, renal vein, and aorta become anatomical landmarks (Figure 3A–E).

### 6.3. Dissection around the Superior Mesenteric Artery

The SMA margin is the most important factor for achieving R0 resection, especially in uncinate process cancer, because the tumor mainly spreads behind the SMA [42,47,48]. It is difficult to understand the anatomy around the SMA during surgery because it is very complex. Recently, region between the SMA and the uncinate process has been called the “mesopancreas” [49,50,51]. Many surgical procedures for complete dissection of the mesopancreas have been reported [41,52,53,54,55,56,57]. However, the range of dissection varies, and the standard dissection range remains unclear. Dense connective tissues exist around the pancreatic head, which is composed of intensive nerve and fibrous tissues (NFTs). It is generally considered that cancer spreads in these areas.

The SMA is covered with NFTs called the SMA nerve plexus. The hard NFTs spread to the uncinate process from the SMA nerve plexus, which is termed as the “pancreatic head plexus II” in the Japanese Classification of Pancreatic Carcinoma [15,43]. Previously, right half-circumferential dissection of the SMA nerve plexus was performed at many facilities [54]. However, extensive dissection of the nerve plexus around the SMA often causes severe diarrhea, which may lead to delays in the induction of adjuvant chemotherapy. Jang et al. conducted a randomized clinical trial comparing extended surgery with right half-circumferential dissection of the SMA nerve plexus and standard surgery without dissection, and revealed that there was no difference in prognosis between the two groups [5]. In their study, the number 14 lymph node was not dissected in the standard group, and the dissection range around the SMA was not clearly described [58,59].

Recently, PD with complete preservation of the SMA nerve plexus has been commonly performed to avoid severe postoperative diarrhea. However, no criteria have been established to indicate an appropriate dissection range for achieving R0 resection in PD with preservation of the SMA nerve plexus. The inferior pancreaticoduodenal artery (IPDA) becomes a good anatomical landmark during the dissection around the SMA [60,61,62,63]. The IPDA forms a common trunk with the first jejunal artery in most cases (J1A) [61,64]. The dissection range can be determined during surgery based on the path of this artery. Various approaches using the IPDA, J1A, and their common arteries as landmarks have been reported for dissection around the SMA [42,56,65,66]. Inoue et al. [42] standardized the anatomical range at levels I–III, depending on the type of tumor, based on the position of the IPDA as an anatomical landmark. They reported that standardizing the dissection range reduced the operative time and blood loss in a study of 162 patients who underwent PD. Of note, the IPDA is covered with intensive NFTs and cannot be identified before initiating the SMA dissection [29,43,49]. In contrast, uncinate process cancer spreads in these intensive NFTs [17,29]. Therefore, alternative anatomical landmarks are needed for the complete dissection of these intensive NFTs in PD with preserving the SMA nerve plexus. Nagakawa et al. [29] evaluated the cancer extension of these areas using pathological specimens from 78 patients who underwent PD for resectable PDAC. According to their results, cancer invasion and/or lymph node metastasis was observed in 14.1% of NFTs (Area C) spreading to the left side of the IPDA root and in 44.9% of NFTs (Area D) spreading between the inferior side of the uncinate process and the posterior side of the jejunal mesentery (Figure 4A–E and Figure 5A–E).

### 6.4. Portal Vein and/or Superior Mesenteric Vein Resection

PV and/or superior mesenteric vein (SMV) resection for patients with PV involvement has been generally accepted with survival benefit of pancreatic cancer [41,67,68,69,70,71,72,73]. The extent of PV infiltration can be diagnosed by preoperative MDCT, and the need for preoperative resection of the PV can be predicted in advance. However, there are cases in which portal vein infiltration is suspected during surgery, even if MDCT does not show tumor infiltration. In addition, it is difficult to distinguish between tumor-related fibrosis and tumor infiltration in the venous wall, and the NCCN guidelines recommend performing PV resections if tumor infiltration is suspected [74].

PDAC often extends to the periphery of the SMV trunk, and the first jejunal vein (J1V) and second jejunal vein (JV) or later branches (J2, 3V) are involved with the tumor. Nevertheless, the resectability of PDAC with JV involvement remains unclear. The NCCN guidelines indicated that “unreconstructible PV/SMV due to tumor involvement or occlusion” is classified as unresectable pancreatic cancer [74]. Some surgeons choose to perform aggressive treatment such as PV/SMV resection with J1V and J2, 3V resection in patients with PDAC [75,76]. However, since the JV is thin, there is concern about the risk of complications, such as portal vein stenosis after portal vein reconstruction [77,78]. Additionally, the survival benefit of PV/SMV resection with JV resection remains unclear. Therefore, it is necessary to clarify the surgical safety and survival benefits of PV/SMV resection with JV resection (Figure 6).

Several running patterns of the J1V have been reported. In 74–99% of J1Vs, the JV flows out from the dorsal side of the SMV, branches off several IPDVs along the uncinate process, passes through the dorsal side of the superior mesenteric artery, and extends to the jejunal mesentery [56,75,79,80]. It is also termed the proximal dorsal JV (PDJV) [56,75]. As the PDJV is in contact with the uncinate process, some surgeons routinely resect the PDJV without reconstruction to ensure a surgical margin, even if combined PV/SMV resection is not required [75,76] (Figure 5).

## 7. Conclusions

The role of surgery has changed dramatically in the current treatment, where multimodal treatment has become important to improve the prognosis of resectable PDAC. Now that effective preoperative and postoperative chemotherapy has been established, it may be necessary to reconsider the areas treated with chemotherapy and the areas treated with surgery. On the other hand, many studies have described that R0 resection is needed even in patients receiving adjuvant therapy. The appropriate dissection range for R0 resection can be simulated preoperatively with MDCT imaging. Therefore, surgeons need to perform a more accurate dissection, balancing both R0 resection and the introduction of adjuvant therapy, based on the precise anatomy.

## Figures and Tables

**Figure 1 cancers-13-03605-f001:**
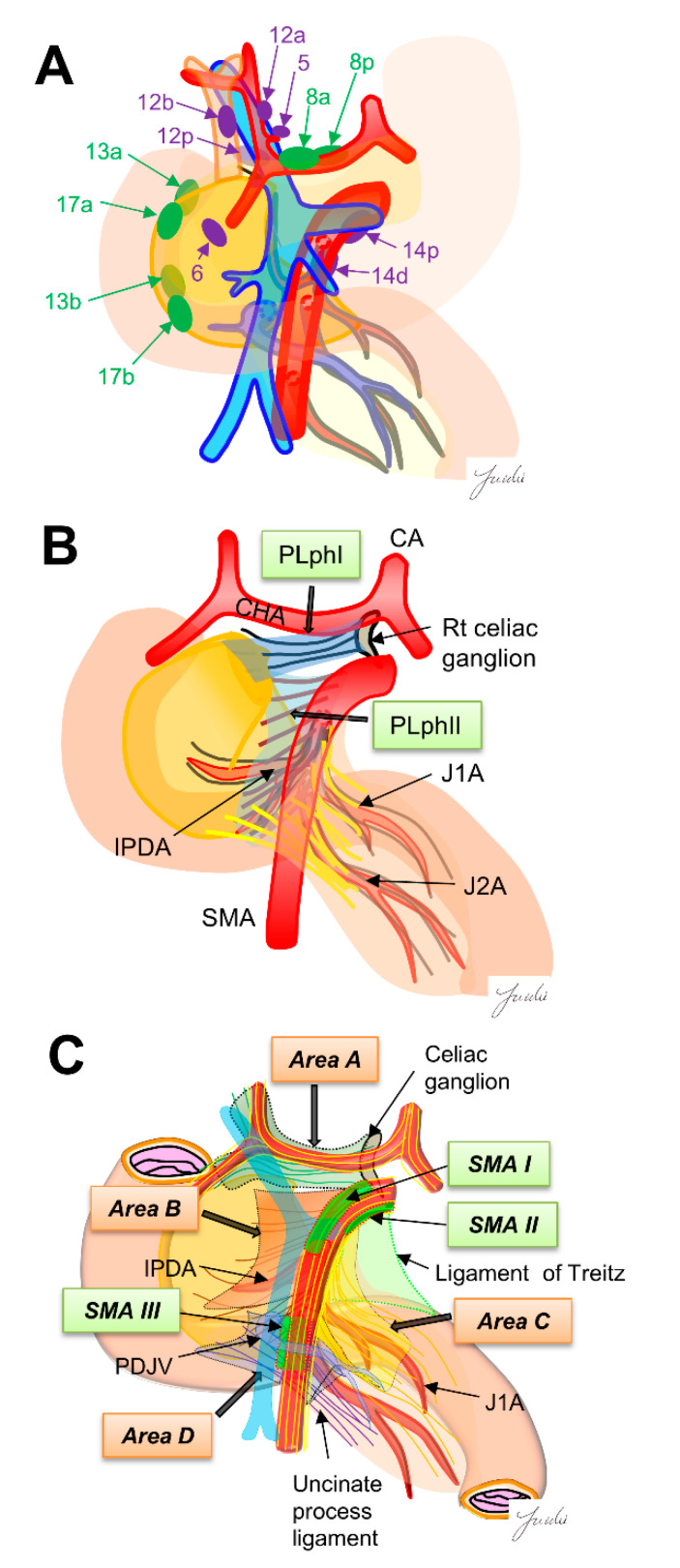
(**A**) The regional lymph nodes are numbered according to the Japanese Pancreatic Cancer classification. Green, D1 region, purple, D2 region. (**B**) Extra-pancreatic nerve plexus around the SMA nerve plexus in the Japanese pancreatic cancer classification. (**C**) Intensive NFTs spreading around the SMA are classified into four areas. Nagakawa et al. [16] classified “intensive NFTs” around the pancreatic head into areas A–D. They also found the three SMA regions (SMAI-III) that can be easily exposed. These regions become anatomical landmarks as “dissection-guiding points” to uniformly dissect each area A–D. PLphI, pancreatic head nerve plexus I; PLphII, pancreatic head nerve plexus II; CA, celiac artery; CHA, common hepatic artery; SMA, superior mesenteric artery; IPDA, inferior pancreaticoduodenal artery; J1A, first jejunal artery; J2A, second jejunal artery; PDJV, proximal dorsal jejunal vein.

**Figure 2 cancers-13-03605-f002:**
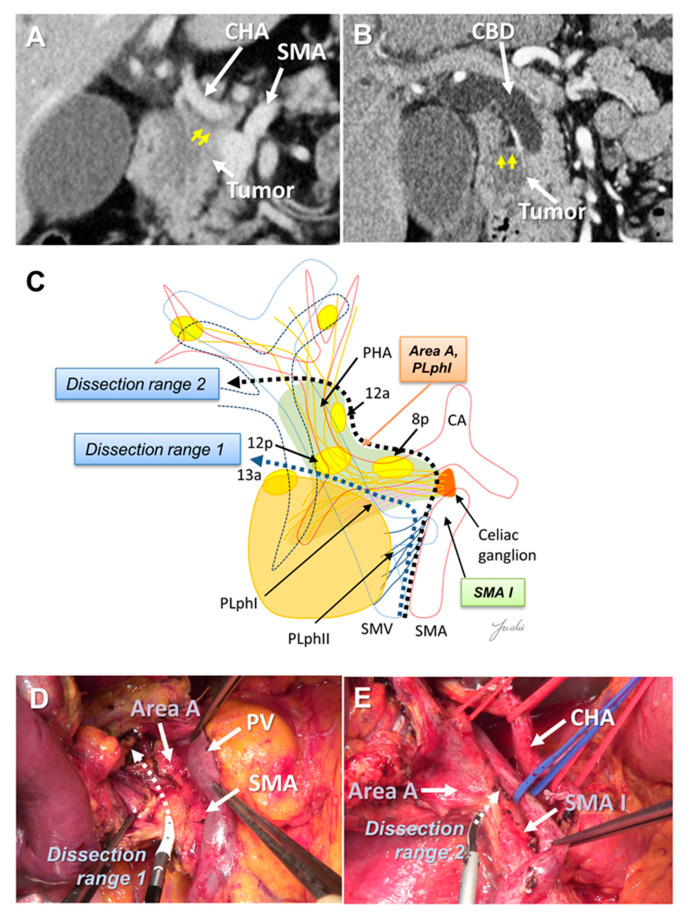
(**A**) Tumor extension is observed near the CHA root and SMA root on the preoperative MDCT findings. (**B**) Tumor extension is observed near the hepatoduodenal ligament. (**C**) Determination of the dissection range based on the MDCT findings. Dissection ranges 1 and 2 can be selected based on the anatomical structure, depending on tumor extension toward to the hepatoduodenal ligament, CHA root, and SMA root. (**D**) Cutting line for dissection range 1. (**E**) Cutting line for dissection range 2. MDCT: multi-detector computed tomography; CBD: common bile duct; CA: celiac artery; CHA: common hepatic artery; SMA: superior mesenteric artery; SMV: superior mesenteric vein.

**Figure 3 cancers-13-03605-f003:**
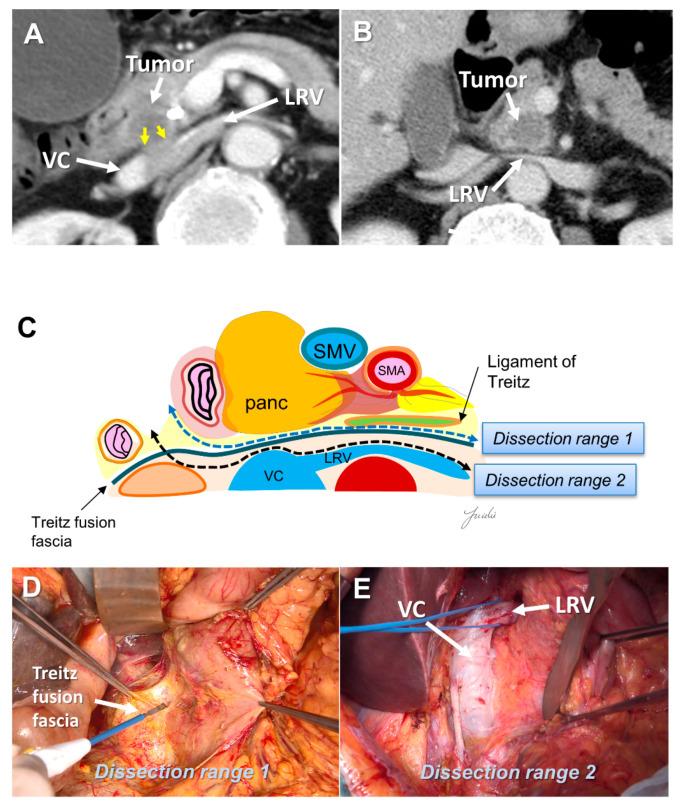
(**A**) Tumor extension to the posterior side of the pancreatic head is observed on the preoperative MDCT findings. (**B**) Tumor extension to the posterior side of the pancreatic head is not observed. (**C**) Determination of the dissection range based on the MDCT findings. Dissection ranges 1 and 2 can be selected based on the anatomical structure, depending on the range of posterior infiltration. (**D**) Surgical findings at dissection range 1. (**E**) Surgical findings at dissection range 2. MDCT: multi-detector computed tomography; VC: vena cava; LRV: left renal vein; SMA: superior mesenteric artery; SMV: superior mesenteric vein.

**Figure 4 cancers-13-03605-f004:**
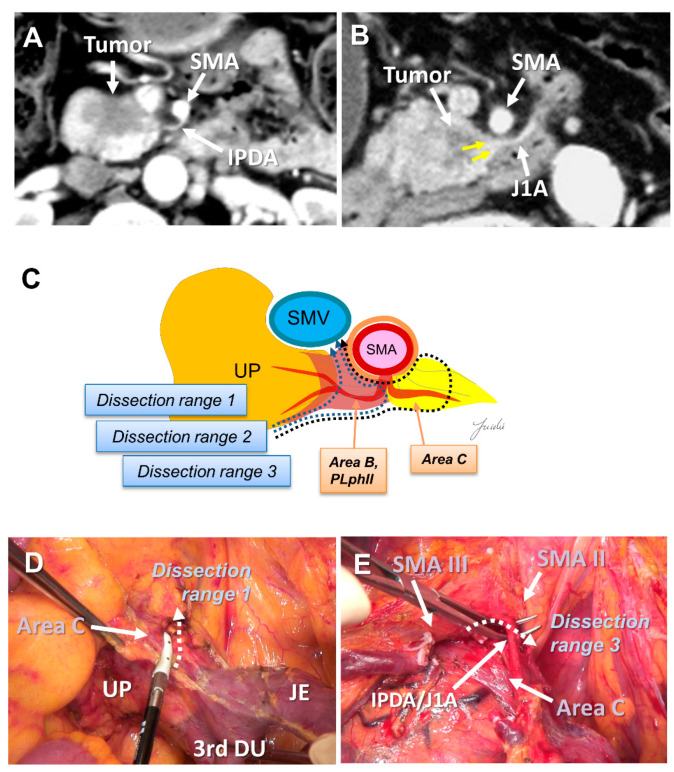
(**A**) Tumor extension to the SMA is not observed on the preoperative MDCT findings. (**B**) Tumor extension to the posterior side of the SMA is observed. (**C**) Determination of the dissection range based on the MDCT findings. Dissection ranges 1, 2, and 3 can be selected based on the anatomical structure, depending on the range of posterior infiltration. (**D**) Cutting line for dissection range 1. (**E**) Cutting line for dissection range 3. MDCT: multi-detector computed tomography; SMA: superior mesenteric artery; IPDA: inferior pancreaticoduodenal artery; J1A: first jejunal artery; UP: uncinate process; 3rd DU: third portion of duodenum; JE: jejunum.

**Figure 5 cancers-13-03605-f005:**
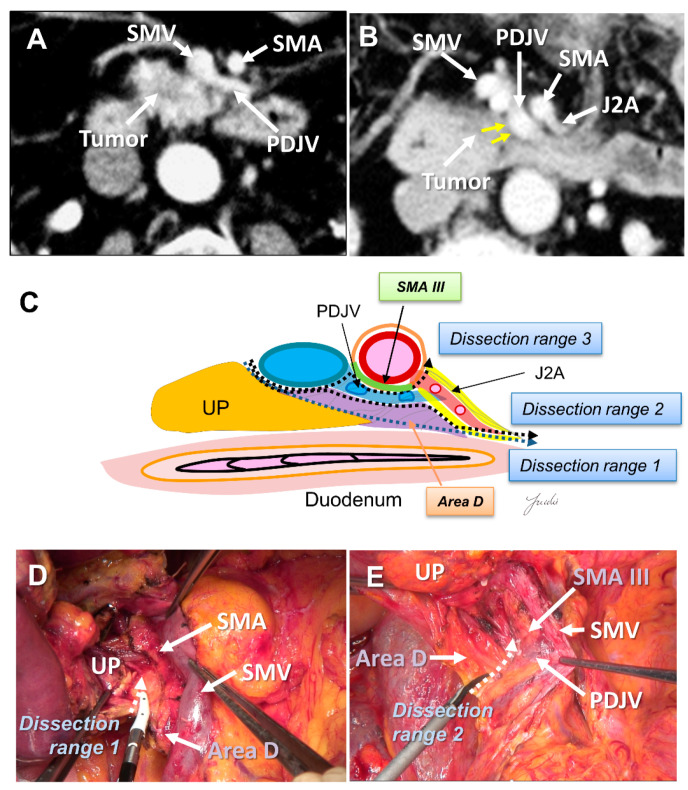
(**A**) Tumor extension on the dorsal side of the jejunal mesentery is not observed on the preoperative MDCT findings. (**B**) Tumor extension on the dorsal side of the jejunal mesentery is observed. (**C**) Determination of the dissection range based on the MDCT findings. Dissection ranges 1, 2, and 3 can be selected based on the anatomical structure, depending on the range of posterior infiltration. (**D**) Cutting line for dissection range 1. (**E**) Cutting line for dissection range 2. MDCT: multi-detector computed tomography; UP: uncinate process; SMA: superior mesenteric artery; SMV: superior mesenteric vein; PDJV: proximal dorsal jejunal vein; J2A: second jejunal artery.

**Figure 6 cancers-13-03605-f006:**
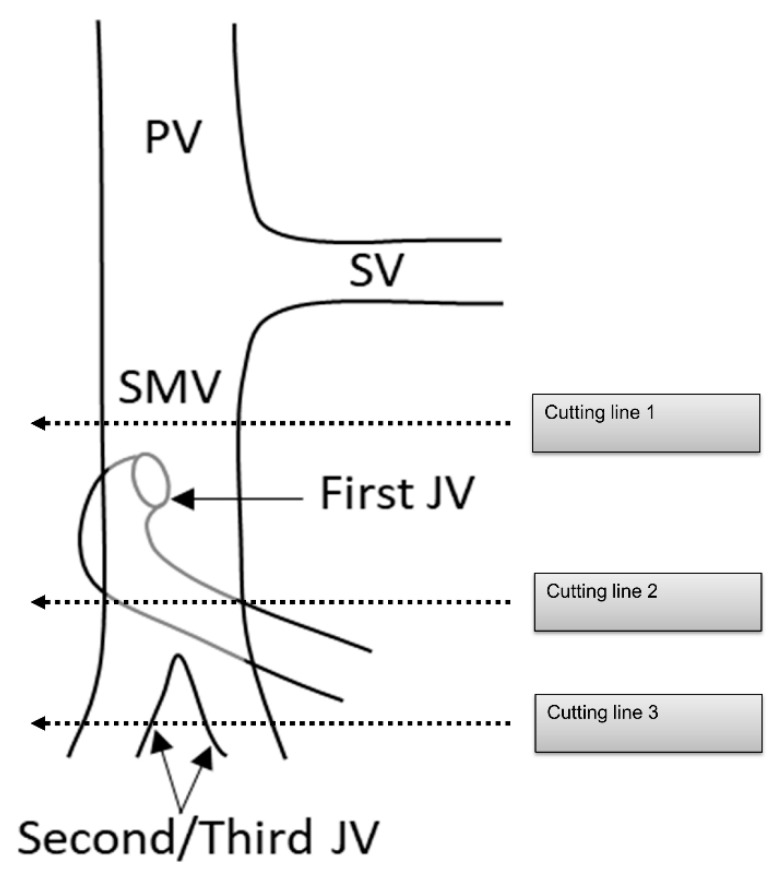
Cutting line for portal vein, superior mesenteric vein, and jejunal vein resection. PV: portal vein; SV: splenic vein; SMV: superior mesenteric vein; JV: jejunal vein.

**Table 1 cancers-13-03605-t001:** Dissection area in randomized controlled trials of extended lymph node dissection and standard dissection for pancreatic head cancer.

Author	Year	Country	Number of Cases	Standard Dissection	Extended Dissection	Standard Dissection	Extended Dissection	Prognosis
				Lymph Node Dissection *		SMA Nerve Plexus Dissection		
Pedrazzoli S et al. [1]	1998	Italy	81	5, 6, 12b, 13, 17	5, 6, 9, 12b, 13, 14, 17, 16a2, 16b1	Not described		MST
								Standard: 335 days
								Extended: 500 days
Yeo C et al. [2]	2002	United States	299	12b2, 12c, 13, 14b, 14v, 17	3, 4, 5, 6, 9, 12b2, 12c, 13, 14b, 14v, 16a2, 16b1, 17	Not described		5-year survival rate
								Standard: 23%
								Extended: 29%
Farnell M et al. [3]	2005	United States	132	3, 4, 6, 8a, 12b1, 12b2, 12c, 13a, 13b, 14a, 14b, 17a, 17b	3, 4, 6, 8a, 8p, 9, 12a1, 12a2, 12b1, 12b2, 12p1, 12p2, 12c, 13a, 13b, 14a, 14b, 14c, 14d, 14v, 16a2, 16b, 17a, 17b	Not described		5-year survival rate
								Standard: 17%
								Extended: 16%
Nimura Y et al. [4]	2012	Japan	112	13a, 13b, 17a, 17b	8a, 8p, 9, 14p, 1416a2, 16b112a, 12b, 12p	None	full circumference dissection	5-year survival rate
								Standard: 15.7%
								Extended: 6.0%
Jang JY et al. [5]	2014	Korea	244	12c, 13, 17	9, 12, 13, 14, 16, 17	None	right half-circumferential dissection	5-year survival rate
								Standard: 44.5%
								Extended: 35.7%

*: Lymph node numbers are listed according to the Japanese Pancreatic Cancer classification. MST: median survival time.

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
