# Peer review of "Reconsideration of the Appropriate Dissection Range Based on Japanese Anatomical Classification for Resectable Pancreatic Head Cancer in the Era of Multimodal Treatment"

_cancers, 2021, doi:10.3390/cancers13143605_

Round 1

Reviewer 1 Report

In the manuscript "Evaluation of the appropriate dissection range for resectable pancreatic head cancer in the era of multimodal treatment" by Yuichi Nagakawa et al., the authors address and review a hot topic in pancreatic cancer. The paper aims to give a comprehensive, relevant and balanced view on the field. Interestingly, the authors present us with very relevant and well-built figures and accompanying diagrams. However, there are several aspects that merit the attention of the authors:

  1. The information reported in the paper is mainly based in the Japanese context and Japanese classification, which does not necessarily apply to the rest of the world. Such detail should be somehow included in the paper title.
  2. There is intense discussion about surgical margins, with several references to R0 and R1 margins, but the authors do not mention in any part of the text the definitions of R0 and R1 that are being followed.
  3. The authors never address the influence of histologic subtype in the patient management.
  4. The authors do not address the issue of pre-surgical biopsy in patient treatment decision.
  5. The authors never mention the issue of nerve involvement in surgical approaches.
  6. The conclusion needs to be reformulated, since it is inadequate in light of all the information discussed in the bulk text. For example, the authors mention blood loss as a prognostic factor in pancreatic cancer, but that issue is scarcely addressed in the bulk text.
  7. Generally, there are some parts of the text that are confusing to follow. The authors should more clearly organize and state their ideas.

Therefore, I consider that this manuscript is interesting and might be relevant for publication in Cancers after correction of all the aspects highlighted above.

Author Response

Reviewer 1

In the manuscript "Evaluation of the appropriate dissection range for resectable pancreatic head cancer in the era of multimodal treatment" by Yuichi Nagakawa et al., the authors address and review a hot topic in pancreatic cancer. The paper aims to give a comprehensive, relevant and balanced view on the field. Interestingly, the authors present us with very relevant and well-built figures and accompanying diagrams. However, there are several aspects that merit the attention of the authors:

Response

Thank you for your warm and positive feedback on our manuscript. We have incorporated changes that reflect your suggestions as described below.

  1. The information reported in the paper is mainly based in the Japanese context and Japanese classification, which does not necessarily apply to the rest of the world. Such detail should be somehow included in the paper title.

Response

We appreciate your suggestion. The main reason for using the Japanese classification in this review is that lymph node classification is internationally used including several RCTs and international consensus meeting (ISGPFs).

Tol JA, Gouma DJ, Bassi C, et al. Definition of a standard lymphadenectomy in surgery for pancreatic ductal adenocarcinoma: a consensus statement by the International Study Group on Pancreatic Surgery (ISGPS). Surgery 2014;156:591-600.

However, as the reviewer pointed out, the title was inappropriate, so we corrected the title as follows.

[Page 1 lines 4]

Reconsideration of the appropriate dissection range based on Japanese anatomical classification for resectable pancreatic head cancer in the era of multimodal treatment                                                                                                            

  1. There is intense discussion about surgical margins, with several references to R0 and R1 margins, but the authors do not mention in any part of the text the definitions of R0 and R1 that are being followed.

Response

We appreciate your suggestion. Accordingly, we have added the following sentence in our revised manuscript:

[Page 6 lines 152-155]

Two definitions have been reported in assessing R1 [1]. American and Japanese classification define R1 as s direct microscopic involvement at resection margin (0 mm rule) [2, 14], and the Royal University of Pathologists classification defines R1 as the presence of cancer cell within 1 mm of resection margin (1mm rule) [3]. It is still unclear which classification reflects the prognosis [4].

  1. The authors never address the influence of histologic subtype in the patient management.

Response

We appreciate your suggestion. However, we could not find any reports on histologic subtypes associated with determining the dissection range in resectable pancreatic cancer in the literature search. Therefore, we could not describe the issue in this review. Basically, this review focuses only on pancreatic ductal adenocarcinoma. To emphasize this, we added the following sentence at the beginning of the introduction section in revised manuscript.

[Page 3 lines 46-48]

Pancreatic ductal adenocarcinoma (PDAC) is recognized as one of the poorest prognostic tumors. Resection is the only treatment that can be expected to long-term survival.

  1. The authors do not address the issue of pre-surgical biopsy in patient treatment decision.

Response

We appreciate your suggestion. However, we could not find any reports which describe pre-surgical biopsy in considering the dissection range in resectable pancreatic cancer in literature search. However, we found some studies regarding dissection range based on pathological diagnosis using intraoperative frozen sections, and described the issues as follows.

[Page 7 lines 188-194]

Intraoperative pathological diagnosis using frozen section is generally performed to determine the pancreatic cutting line to avoid positive pancreatic neck margins. Addition-ally, respectability status is also evaluated using frozen section of the SMA margin in some facilities. Nirsgke et al. reported that long-term survival was improved by re-resecting the positive surgical margin found using frozen section to achieve R0 resection [30]. However, many studies reported that intraoperative frozen section-based re-resection of R1 margin does not improve overall survival for PDAC patients [31-33].

  1. The authors never mention the issue of nerve involvement in surgical approaches.

Response

We appreciate your suggestion. Accordingly, we have added the following sentence in our revised manuscript:

[Page 6 lines 164- Page 7 line 169]

It has been reported that dissection of the NFTs around the pancreatic head is important for achieving R0 resection, because PDAC often infiltrate these NFTs [27, 28]. However, the appropriate dissection range of NFTs has not been fully discussed, and a common classi-fication showing the anatomical structure of NFTs around the pancreatic head is needed to determine the dissection range. Several classifications have been shown for the anato-my of NFTs.

  1. The conclusion needs to be reformulated, since it is inadequate in light of all the information discussed in the bulk text. For example, the authors mention blood loss as a prognostic factor in pancreatic cancer, but that issue is scarcely addressed in the bulk text.

Response

We appreciate your suggestion. As the reviewer's advice, it was not appropriate to mention other issues which was not reviewed in the conclusion. Therefore, we removed all issues which is not address in the text. Additionally, we completely modified the following sentence in conclusion session.

[Page 17 lines 561-569]

The role of surgery has changed dramatically in the current treatment, where multi-modal treatment has become important to improve the prognosis of resectable PDAC. Now that effective preoperative and postoperative chemotherapy has been established, it may be necessary to reconsider the areas treated with chemotherapy and the areas treated with surgery. On the other hands, many studies have described that R0 resection is need-ed even in patients receiving adjuvant therapy. Appropriate dissection range for R0 resec-tion can be simulated preoperatively with MDCT imaging. Therefore, surgeons need to perform a more accurate dissection, balancing both R0 resection and the introduction of adjuvant therapy, based on the precise anatomy.

  1. Generally, there are some parts of the text that are confusing to follow. The authors should more clearly organize and state their ideas.

Response

We appreciate your valuable comment and apologize for the lack of explanation. To make it easier to understand, the structure of the text has been revised. The corrected parts are shown in red.

Reviewer 2 Report

The manuscript treats an important issue on the cancer therapy, particularly the pancreatic head cancer. The review summarized great information to discuss the appropriate resection range of pancreatic head cancer. The manuscript has high quality, and all figures are clearly presented and well-discussed. Therefore, the manuscript merits publication. Only one suggestion that the text and the graph indicating lymph nodes in figures can be drawn with different colors to identify Group 1 and Group 2. 

Author Response

Reviewer 2

The manuscript treats an important issue on the cancer therapy, particularly the pancreatic head cancer. The review summarized great information to discuss the appropriate resection range of pancreatic head cancer. The manuscript has high quality, and all figures are clearly presented and well-discussed. Therefore, the manuscript merits publication. Only one suggestion that the text and the graph indicating lymph nodes in figures can be drawn with different colors to identify Group 1 and Group 2. 

Response

Thank you for your warm and positive feedback on our manuscript. We have modified the figure 1 according to reviewer’s suggestion.

Round 2

Reviewer 1 Report

In the revised version of the manuscript "Evaluation of the appropriate dissection range for resectable pancreatic head cancer in the era of multimodal treatment" now entitled “Reconsideration of the appropriate dissection range based on Japanese anatomical classification for resectable pancreatic head cancer in the era of multimodal treatment” by Yuichi Nagakawa et al., the authors address and review a hot topic in pancreatic cancer. The paper aims to give a comprehensive, relevant and balanced view on the field. Interestingly, the authors present us with very relevant and well-built figures and accompanying diagrams. Finally, the authors addressed well most of the aspects raised following the initial review of the paper. Thus, I consider that this manuscript is interesting and relevant for publication in Cancers in the present format.